# Antimicrobial Agent Use for Urinary Tract Infection in Long-Term Care Facilities in Spain: Results from a Retrospective Analytical Cohort Analysis

**DOI:** 10.3390/antibiotics13020152

**Published:** 2024-02-03

**Authors:** Priscila Matovelle, Bárbara Olivan-Blázquez, Rosa Magallón-Botaya, Ana García-Sangenís, Ramon Monfà, Rosa Morros, Alicia Navarro Sanmartín, Jesús Mateos-Nozal, Carmen Sáez Bejar, Consuelo Rodríguez Jiménez, Elena López Pérez, Carl Llor

**Affiliations:** 1Geriatrics Department, Hospital San Juan de Dios, 50006 Zaragoza, Spain; priscilamatovelle@gmail.com; 2Geriatrics Department, Universidad de Zaragoza, 50009 Zaragoza, Spain; 3Group B21-23R, Health Research Institute of Aragon (IISA), 50009 Zaragoza, Spain; rosamaga@unizar.es; 4Network for Research on Chronicity, Primary Care and Health Promotion (RICAPPS, RD21/0016/0005), 50015 Zaragoza, Spain; 5Department of Psychology and Sociology, University of Zaragoza, 50009 Zaragoza, Spain; 6Medicine Department, Universidad de Zaragoza, 50009 Zaragoza, Spain; 7Fundació Institut Universitari per a la Recerca a l’Atenció Primària de Salut Jordi Gol i Gurina, 08007 Barcelona, Spain; agarcia@idiapjgol.org (A.G.-S.); rmonfa@idiapjgol.info (R.M.); rmorros@idiapjgol.org (R.M.); carles.llor@gmail.com (C.L.); 8Pharmacology Department, Universitat Autònoma de Barcelona, Bellaterra (Cerdanyola del Vallès), 08193 Bellaterra, Spain; 9CIBER en Enfermedades Infecciosas Instituto Carlos III, 28029 Madrid, Spain; 10Nursing Home, Aragon Institute of Social Services (IASS), 50540 Borja, Spain; anavarros@aragon.es; 11Geriatrics Department, Hospital Universitario Ramón y Cajal (IRYCIS), 28034 Madrid, Spain; jesus.mateosdel@salud.madrid.org; 12Internal Medicine Department, Hospital Universitario La Princesa, Instituto de Investigación Sanitaria (IIS-Princesa), 28006 Madrid, Spain; carmenmaria.saez@salud.madrid.org; 13Research Institute of Princesa (IIS Princesa), 28006 Madrid, Spain; 14Pharmacology Department, Complejo Hospitalario Universitario de Canarias, 38320 Santa Cruz de Tenerife, Spain; conrodjim@gmail.com; 15Institut Català de la Salut, 08038 Barcelona, Spain; elopezp.apms.ics@gencat.cat; 16Research Unit for General Practice, Department of Public Health, University of Southern Denmark, 5230 Odense, Denmark

**Keywords:** elderly population, nursing home, urinary tract infection, antibiotic prescriptions

## Abstract

Urinary tract infections (UTIs) are highly prevalent in long-term care facilities, constituting the most common infection in this setting. Our research focuses on analyzing clinical characteristics and antimicrobial prescriptions for UTIs in residents across nursing homes (NH) in Spain. This is a retrospective analytical cohort analysis using a multifaceted approach based on the normalization process theory to improve healthcare quality provided by nursing staff in 34 NHs in Spain. In this study, we present the results of the first audit including 719 UTI cases collected between February and April 2023, with an average age of 85.5 years and 74.5% being women. Cystitis and pyelonephritis presented distinct symptom patterns. Notably, 6% of asymptomatic bacteriuria cases were treated. The prevalence of dipstick usage was 83%, and that of urine culture was only 16%, raising concerns about overreliance, including in the 46 asymptomatic cases, leading to potential overdiagnosis and antibiotic overtreatment. Improved diagnostic criteria and personalized strategies are crucial for UTI management in NHs, emphasizing the need for personalized guidelines on the management of UTIs to mitigate indiscriminate antibiotic use in asymptomatic cases.

## 1. Introduction

In recent decades, Europe has experienced a significant increase in the number of geriatric individuals. The population projections of the European Union estimate that the old-age dependency ratio will increase from 33.0% in 2022 to 59.7% in 2100 [1].

Demographic changes are transforming the healthcare landscape. Older adults represent a larger proportion of the population, increasing the demand for long-term care facilities. Nursing homes (NHs) are intended to mimic the home environment of residents, but they often become spaces where people with significant underlying conditions live in limited spaces, sharing caregivers in a communal setting [2]. Despite providing less intensive medical care, long-term care facilities have not been immune to healthcare-associated infections, affecting the vulnerable NH population [3,4].

Infections represent a major challenge for older people living in NHs, as the compromised condition of their immune system known as immunosenescence and underlying health problems make them more susceptible to these conditions [5].

Urinary tract infections (UTIs) are common infections among elderly people living in NHs in Europe [6]. This pathology ranges from asymptomatic bacteriuria to UTI-associated sepsis requiring hospitalization. The diagnosis of a symptomatic UTI in older adults usually requires the presence of localized genitourinary symptoms, fever, pyuria, and a urine culture confirming the presence of a uropathogen [7]. In cases of symptomatic UTIs, antimicrobial therapy is considered appropriate [8].

Given the potential for UTIs to lead to severe infections, healthcare professionals may diagnose UTIs relying solely on vague symptoms, such as changes in behavior or alterations in the appearance of urine.

Antibiotics are often prescribed for this infection, even though a considerable portion of these prescriptions are considered inappropriate [9,10,11]. This practice contributes significantly to increasing antimicrobial resistance (AMR), which poses a threat to public health, healthcare systems, and social well-being [12]. Infections caused by antibiotic-resistant bacteria are associated with increased morbidity and mortality, as well as increased treatment costs due to the high risk of complications and hospital admissions [13,14].

AMR is linked to significant economic costs. In Europe, for instance, AMR is estimated to result in an annual economic burden surpassing EUR 9 billion [15,16]. In NHs, this concern is exacerbated by the problem of suboptimal antimicrobial prescribing due to a lack of diagnostic capacity [17]. The aging of the population in NHs has presented a representative change with an increase in complexity, including higher rates of multimorbidity and frailty [18]. This demographic change leads to increased susceptibility to serious complications, especially infectious diseases, highlighting the need for improved antimicrobial stewardship programs and prevention measures for healthcare-associated infections [2].

Therefore, the aim of our study is to evaluate the clinical characteristics and prevalence of antimicrobial prescriptions for UTIs in geriatric residents residing in NHs throughout Spain.

## 2. Results

### 2.1. Number of Registrations

Between February and April 2023, our study included 1505 infection registries in 34 NHs across Spain in five different nodes. During this period, healthcare workers in these facilities documented a total of 719 suspected cases of UTIs in elderly patients. The mean age was 85.5 (SD: 7.9) years; 74.5% were women. Additionally, 42 (5.8%) patients had urinary catheters.

The most common symptoms reported included confusion, foul-smelling urine, and cloudy urine. The most common diagnosis was cystitis. A urine dipstick test was conducted for a significant percentage of the patients, specifically 83.6%. Among those tested, 40% were prescribed fosfomycin as part of their treatment (Table 1).

As shown in Figure 1, when cross-referencing variables between general symptoms and those specific to UTIs, the most prominent circles converged at the intersection of confusion with foul-smelling and cloudy urine, indicating the highest frequency of recorded symptoms. These numbers represent exact numbers of patients.

A total of 46 residents who exhibited neither general nor specific symptoms indicative of a UTI were reported, as they were prescribed antibiotics for this infection. Within this cohort, 43 persons (93.5%) received antibiotic treatment, predominantly with fosfomycin for 1 or 2 days. Out of the 43 treated cases, urine dipstick diagnostic tests were conducted in 65% of instances, while urine cultures were carried out in 35%.

### 2.2. Diagnoses

The two primary diagnoses of UTIs presented distinct symptom patterns. Notably, there was a similarity in the percentage of symptoms between cystitis and undetermined infection, suggesting that the cases in which the diagnosis “none of the above” was selected probably corresponded to cases of cystitis. In contrast, the symptomatic pattern of patients diagnosed with pyelonephritis diverged from the other two groups. Patients with pyelonephritis had a higher prevalence of dysuria and gross hematuria and, surprisingly, a lower presence of confusion compared to cases of cystitis and undetermined infection. No statistically significant differences were observed among the three groups, likely due to the limited number of pyelonephritis cases (n = 14). However, when comparing cystitis with the absence of both cystitis and pyelonephritis, a strong correlation is evident (R2 = 0.94), indicating that the symptom percentages for these two diagnoses are nearly identical (Figure 2).

In the examination and diagnosis of residents with and without urinary catheterization, both groups showed a similar distribution of genders. Interestingly, the prevalence of undetermined infections was notably higher among residents with urinary catheterization. Despite the similarity in the most frequent diagnosis between the two groups, this disparity in the incidence of undetermined infections suggests a possible association with the presence of urinary catheters (Table 2).

### 2.3. Antibiotic Therapy

All included cases received antibiotic monotherapy. A total of 40 residents (5.5%) were not treated with antibiotics. Interestingly, the untreated group had slightly fewer cases of confusion, cloudy urine, and foul-smelling urine compared to the treated residents. In addition, there was a slightly increased prevalence of no symptoms in the untreated group, although the differences between the two groups were minimal. No statistically significant differences were observed among the three groups, likely due to the limited number of pyelonephritis cases (Figure 3).

Concerning the diagnosis, cystitis was the most frequently recorded, accounting for almost 70% of the patients; however, a urine culture was conducted for only 15% of the patients. Unfortunately, results on the isolated microorganisms and their resistance profile were not available (Table 3).

When studying antibiotic prescriptions, differences were identified in the prescribing patterns for pyelonephritis in comparison to acute cystitis. Specifically, there was a higher prevalence of quinolones and cephalosporins in pyelonephritis cases, while fosfomycin was less frequently prescribed compared to cases of acute cystitis. No statistically significant differences were observed among the three groups.

A total of 353 women without an indwelling catheter were diagnosed with uncomplicated cystitis and treated with antibiotics. Nearly half of these treatments corresponded to fosfomycin, with cephalosporins, quinolones, and amoxicillin and clavulanic acid following in frequency. The median treatment duration was observed to be 7 days. For instance, the two-day fosfomycin regimen was more commonly used than the single-dose regimen, and nitrofurantoin was predominantly given for 7 days. Similarly, a substantial number of quinolones and cephalosporins were prescribed for a duration of 7 days. Furthermore, a notable percentage of residents received antibiotic regimens lasting 10 days or more (Table 4).

## 3. Discussion

We found a prevalent pattern in which general symptoms, especially confusion, converged with specific symptoms, such as foul-smelling urine and cloudy urine as the most frequently recorded manifestations in cases with a diagnosis of cystitis. In contrast, patients diagnosed with pyelonephritis presented a distinct symptom profile, characterized by a higher prevalence of dysuria and gross hematuria and a lower presence of confusion compared to cases of cystitis and indeterminate infection. However, 6% of residents showed no general or specific symptoms, and most of them were treated with antibiotics. Another remarkable finding was the high prescription of cephalosporins and fluoroquinolones and the high use of long therapies, which are not in line with the current clinical guidelines.

### 3.1. Limitations

This study has several limitations. Participants participated on a voluntary basis, and as shown in some studies, volunteer professionals might have a greater interest in quality improvement programs and research than the general population of professionals [19]. From a theoretical perspective, the treatment decision ideally follows the diagnosis decision-making. Diagnostic procedures and treatment decisions are closely intertwined. Healthcare professionals may determine an antibiotic prescription concurrently with or even prior to definitively diagnosing the patient’s condition. Subsequently, participants might adapt the diagnosis to align with the treatment decision, potentially introducing a diagnostic misclassification bias [20].

### 3.2. Interpretation of Results and Comparison with Literature

The most prevalent symptoms in the current study were confusion, foul-smelling urine, and cloudy urine. This contrasts with a cohort study that identified dysuria, change in urine characteristics, and change in mental status as significantly associated with the combined outcome of bacteriuria plus pyuria [21]. Yet, in another study, the most common symptoms were fever without localized genitourinary symptoms or signs [22]. Caterino et al. established a correlation between age and UTI symptoms, revealing that adults aged ≥ 65 diagnosed with UTIs in the emergency department lacked fever and typical urinary tract symptoms. Notably, those aged ≥ 85 are more likely to manifest altered mental status. Additionally, NH residents are characterized by a reduced likelihood of urinary tract symptoms but an increased likelihood of fever and altered mental status [23].

Nevertheless, a challenge arises as some residents presenting with a positive urine culture and altered mental status, falls, lack of appetite, or compromised mobility are often diagnosed with a UTI. This trend, observed in various NHs, highlights the importance of refining diagnostic criteria to ensure accurate identification of UTIs and mitigate the risk of unnecessary antibiotic treatment [21]. Hence, McGeer’s criteria provide a systematic framework for diagnosing UTIs in both non-catheterized and catheterized residents. These criteria offer a comprehensive and standardized approach to guide healthcare professionals in accurately diagnosing UTIs. It is important to treat catheterized residents only if they have a symptomatic infection [24]. However, there is a debate over the utilization of these criteria in patients with advanced dementia, where antibiotic overuse is notably prevalent. A study focusing on advanced dementia patients in NHs revealed that merely 19% of treated probable UTIs aligned with these diagnostic criteria [25]. Other research indicated that despite the administration of antibiotics for suspected UTIs in institutionalized patients with severe dementia, there was no notable improvement in survival outcomes [26,27].

As for the 46 patients who were asymptomatic and were treated with antibiotics, there is limited direct evidence concerning cases where antibiotics are administered specifically to asymptomatic patients within this population group. It is crucial for future research to concentrate on comprehending the factors contributing to antibiotic administration despite the absence of symptoms in patients. 

Recognizing that diagnosing UTIs in the geriatric population should not rely solely on signs and symptoms, bacteriological urine culture stands out as the standard diagnostic test for UTIs [26]. Another frequently employed diagnostic tool is the urine dipstick. According to a meta-analysis, urine dipstick tests prove beneficial primarily in ruling out the presence of an infection when both nitrites and leukocyte-esterase yield negative results [28]. Furthermore, a notable proportion of older adults exhibit asymptomatic bacteriuria (AB), defined as >100,000 colony-forming units on urine culture, without specific urinary tract symptoms [29]. The prevalence of bacteriuria varies significantly, ranging from 25% to 50% in institutionalized women and from 15% to 40% in men [8,30]. A Belgian study further supported this observation, revealing a high prevalence, reaching 80–90%, of AB in a specific group of female NH residents characterized by urinary incontinence or a high degree of dependence and disorientation [31]. Although AB may act as a protective barrier to symptomatic UTIs, antibiotic therapy is generally not indicated as it is unnecessary and even harmful [7,8,32].

In our study, we observed that urine culture was conducted in a relatively low proportion, accounting for only 16% of the patients, while the dipstick method was used in 83% of the reported cases. The prevalent use of dipstick testing may be influenced by factors like ease of implementation, rapid results, and practical considerations, which are crucial in long-term care facilities where efficient diagnostic approaches are crucial [33]. It is remarkable that only 16% of the records requested urine cultures with antibiograms, indicating a relatively low rate of thorough microbiological investigation. Despite this, there was substantial empirical antibiotic use, highlighting a significant gap between diagnostic testing practices and antibiotic prescribing prevalence. This raises concerns about indiscriminate antibiotic use without a clear understanding of the microbial landscape. 

A total of 94.4% of the records involved the prescription of antibiotics. NHs are particularly notable for their elevated rates of antibiotic use, with estimates suggesting that as much as 75% of prescribed antibiotics may be inappropriate [34,35]. The most prevalent antibiotics prescribed in our study were fosfomycin, accounting for 40% of all the cases, followed by cephalosporins and fluoroquinolones. In this line, Pulia et al. found that fluoroquinolones constituted the most prescribed class of antibiotics for UTIs, with a frequency of 35.6% [35]. While nitrofurantoin and trimethoprim took precedence in a study with a prevalence of 39% and 41%, respectively [36], fluoroquinolones, sulfonamides, and first-generation cephalosporins were noted as the most prescribed antibiotics in another study [37]. 

We analyzed the time of antibiotic treatment given in women with simple cystitis without catheterization. The median treatment duration was observed to be 7 days. For instance, the two-day fosfomycin regimen was more commonly used than the single-dose regimen. Substantial numbers of quinolones, nitrofurantoin, and cephalosporins were prescribed for a duration of 7 days. Furthermore, a notable percentage of residents received antibiotic regimens lasting 10 days or more. In this line, in a meta-analysis of 9605 adult non-pregnant women with uncomplicated cystitis, comparing 3-day oral antibiotic therapy with prolonged therapy lasting 5 days or more, no significant difference in symptomatic failure rates was found between the two regimens, both in short-term (RR 1.16, 95% CI: 0.96–1.41) and long-term follow-up (RR 1.17, 95% CI: 0.99–1.38). However, the 3-day treatment was less effective in preventing bacteriological failure, with a relative risk of 1.37 (95% CI: 1.07–1.74) for short-term follow-up and 1.47 (95% CI: 1.22–1.77) for long-term follow-up. Importantly, adverse effects were more common in the prolonged therapy group, indicating a relative risk of 0.83 (95% CI: 0.79–0.91) [38].

Previous investigations conducted in NHs have disclosed that, generally, the decision to initiate antibiotic therapy was deemed appropriate in 49% to 63% of cases [39,40,41]. The Antimicrobial Consumption in the EU/EEA (ESAC-Net) report reported the prevalence of antimicrobial agents in long-term care facilities across 24 European Union countries during 2016–2017. These results accounted for an overall prevalence of antimicrobial use at 5.8%, with Spain recording the highest consumption rate at 11.7% [42]. Given the high incidence of antibiotic prescription in this population, which can be both ineffective and potentially harmful, the research group recommended a treatment protocol aligned with the guidelines outlined by Gupta et al. and Bonkat et al. [43,44].

The treatment of 43 out of the 46 cases with asymptomatic bacteriuria using antibiotics and the observation that a majority of these patients underwent dipstick analysis emphasize the tendency among professionals to treat patients based on urinary dipstick results even in the absence of specific symptoms. This indicates that NH professionals might rely on diagnostics too much, while not paying enough attention to patients’ symptoms [45].

## 4. Materials and Methods

### 4.1. Design

This was a retrospective analytical cohort analysis. We performed a quality control methodology, which is an evidence-based multifaceted intervention for improving quality of care by implementing guidelines. The proposed multifaceted intervention sought to address all the dimensions of the normalization process theory by facilitating the open discussion of variation in the behavior of the different NHs and strengthening the communication between the nursing staff and the residents. To evaluate the results of the intervention, two point-prevalence audits were performed. These audits were carried out before and after the multifaceted intervention, where the participants registered anonymized data of the consultations they had regarding UTIs in two charts, in two short periods of time: before the intervention and after the intervention. In this study, we describe the characteristics of the registrations of UTIs reported in the first audit in the period from February to April 2023. Thirty-four NHs across Spain were recruited. The participants who documented the infections were doctors, nurses, nurse assistants, or nurse helpers in each nursing home.

### 4.2. Audit Registrations

On the first day, the participants registered the infection audit and counted the percentage of residents receiving antibiotics. The chart included cases of UTI, their general symptoms (fever, chills, confusion, arthromyalgias, or no general symptoms), specific symptoms (dysuria, urinary urgency, frequency, urinary incontinence, back or flank pain, blood in urine, foul-smelling urine, cloudy urine, and no symptoms), diagnosis (cystitis, pyelonephritis, none), and antibiotic treatment (penicillin V, amoxicillin, amoxicillin and clavulanic acid, macrolide or clindamycin, cephalosporin, fosfomycin, nitrofurantoin, trimethoprim +sulfamethoxazole, quinolone, another antibiotic, or none).

### 4.3. Statistical Analisis

We used the chi-square test to the frequencies of prescriptions for each group of professionals and setting/country to test the null hypothesis of no effect of the audit. Antibiotic prescription was considered as the dependent variable (yes/no). The models controlled for symptoms and duration, as well as for the demand for antibiotics by the resident. The goodness of fit was assessed using the Wald test of the model, with the deviance test being used to compare alternative models. Statistical significance was considered with *p* < 0.05. The data were analyzed with the Stata v17 statistical program.

### 4.4. Informed Consent

All professionals who were willing to participate in this study were given written and verbal information (Participant Information Sheet) and had time to consider the information and ask questions. If they agreed to participate, they were asked to complete and sign 2 copies of the Participant Consent Form on paper; one copy was securely stored in the coordinator site, and the other copy was given to the participant. Each professional was identified by a code. The audit templates identified the professionals by these codes.

### 4.5. Ethical Considerations

This study was reviewed and approved by the Ethics Committee Board of the Primary Care Research Institute IDIAP Jordi Gol, Barcelona, Spain (reference number 22/119-P).

## 5. Conclusions

This study highlights the prevalence and challenges of urinary tract infections in NHs, emphasizing the need for refined diagnostic criteria to avoid overdiagnosis and inappropriate antibiotic use. McGeer’s criteria are suggested for systematic UTI diagnosis, although challenges persist in advanced dementia cases. We also discuss the diagnostic tools employed, such as urine culture and dipstick tests, emphasizing the importance of tailored and thoughtful diagnostic strategies, particularly in vulnerable populations. The discrepancy between diagnostic testing practices and empirical antibiotic prescribing raises concerns about potential indiscriminate antibiotic use. This study also reports a high rate of antibiotic prescription, primarily involving fosfomycin, cephalosporins, and fluoroquinolones, echoing concerns about antibiotic resistance. The analysis of antibiotic treatment duration reveals variability, with a substantial number of residents receiving regimens lasting 10 days or more. We conclude by recommending a tailored treatment protocol for UTIs in long-term care, considering factors such as gender, comorbidities, and infection type, while discouraging unnecessary antibiotic use for asymptomatic conditions.

## Figures and Tables

**Figure 1 antibiotics-13-00152-f001:**
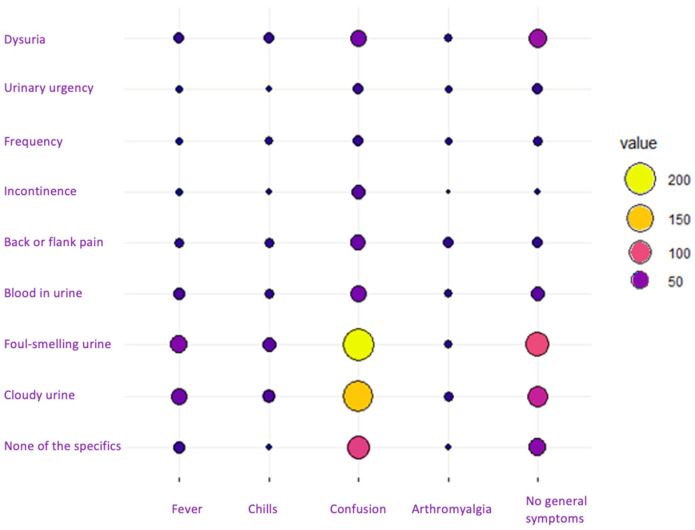
Interaction chart between general symptoms and specific symptoms for urinary tract infections.

**Figure 2 antibiotics-13-00152-f002:**
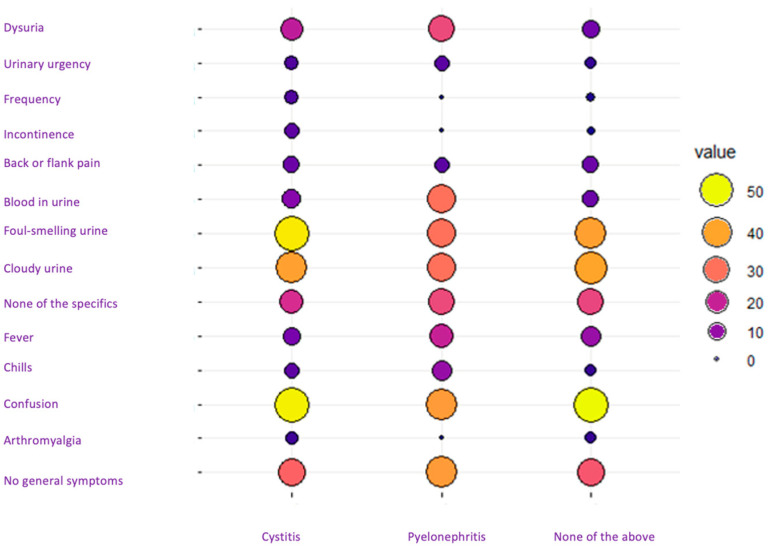
Interaction graph between general and specific symptoms and diagnosis.

**Figure 3 antibiotics-13-00152-f003:**
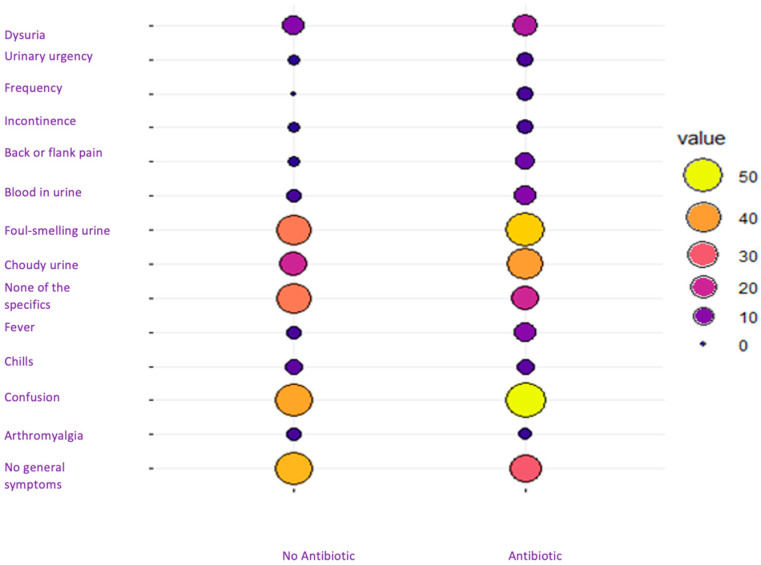
Correlation between symptoms and antibiotic prescription.

**Table 1 antibiotics-13-00152-t001:** Demographics, diagnostic tests, and symptomatology of registered urinary tract infections.

		Total
		n (%)
		719 (100)
Women (%)		536 (74.5)
Men (%)		183 (25.5)
Age, mean (SD)		85.51 (7.93)
Urinary catheter (%)		42 (5.8)
Diagnostic test	Urine dipstick	601 (83.6)
	Urinary culture	118 (16.4)
General signs and symptoms	Fever	88 (12.2)
	Chills	52 (7.2)
	Confusion	400 (55.6)
	Arthromyalgia	26 (3.6)
	No general symptoms	231 (32.1)
Symptoms and signs of urinary tract infection	Dysuria	124 (17.2)
	Urinary urgency	36 (5.0)
	Frequency	34 (4.7)
	Incontinence	37 (5.1)
	Back or flank pain	64 (8.9)
	Gross blood in urine	83 (11.5)
	Foul-smelling urine	356 (49.5)
	Cloudy urine	302 (42.0)
	None of the specifics	168 (23.4)

n, number; %, percentage, SD: standard deviation.

**Table 2 antibiotics-13-00152-t002:** Distribution of different urinary tract infections according to sex and whether they have a urinary catheter.

Diagnosis	Without Urinary Catheter (n = 677)			With Urinary Catheter (n = 42)			
	Men	Women		Men	Women		Total
	(n = 157)	(n = 520)	*p*	(n = 26)	(n = 16)	*p*	(n = 719)
	n (%)	n (%)		n (%)	n (%)		n (%)
Cystitis	101 (64.3)	374 (71.9)	0.1	13 (50.0)	10 (62.5)	0.3	498 (69.3)
Pyelonephritis	3 (1.9)	10 (1.9)	0.1	0 (0.0)	1 (6.25)	0.3	14 (2)
None of the above	35 (22.3)	100 (19.2)	0.1	12 (46.2)	4 (25.0)	0.3	151 (21.0)
Missing	18 (11.5)	36 (6.9)	0.1	1 (3.8)	1 (6.25)	0.3	56 (7.7)

No statistically significant differences were observed.

**Table 3 antibiotics-13-00152-t003:** Diagnosis and treatment according to sex.

		Total	Women	Men	
		n (%)	n (%)	n (%)	
		719 (100)	536 (74.5)	183 (25.5)	*p*
Diagnosis	Cystitis (%)	498 (69.3)	384 (71.6)	114 (62.2)	0.1
	Pyelonephritis (%)	14 (1.9)	11 (2.0)	3 (1.6)	0.7
	None of the above (%)	151 (21.0)	104 (19.4)	47 (25.6)	0.7
Antibiotic dispensed	Penicillin V (%)	0 (0.0)	0 (0.0)	0 (0.0)	0.7
	Amoxicillin (%)	3 (0.4)	2 (0.37)	1 (0.5)	0.7
	Amoxicillin/clavulanic acid (%)	65 (9.0)	45 (8.3)	20 (10.9)	0.3
	Macrolide or clindamycin (%)	1 (0.1)	1 (0.2)	0 (0.0)	0.5
	Cephalosporin (%)	129 (17.9)	92 (17.1)	37 (20.2)	0.3
	Fosfomycin (%)	288 (40.1)	228 (42.5)	60 (32.7)	0.2
	Nitrofurantoin (%)	54 (7.5)	42 (7.8)	12 (6.5)	0.5
	Trimethoprim/sulfamethoxazole (%)	34 (4.7)	19 (3.5)	15 (8.2)	0.1
	Quinolone (%)	86 (12.0)	59 (11.0)	27 (14.7)	0.1
	Other antibiotic (%)	26 (3.6)	19 (3.5)	7 (3.8)	0.8
	No antibiotic (%)	40 (5.6)	32 (5.9)	8 (4.3)	0.4

No statistically significant differences were observed.

**Table 4 antibiotics-13-00152-t004:** Days of antibiotic treatment given in women with simple cystitis without catheterization.

	Total No. Antibiotic Prescriptions	Duration of Antibiotic Treatment (days)									
		1	2	3	5	6	7	8	9	10	>10
		n (%)	n (%)	n (%)	n (%)	n (%)	n (%)	n (%)	n (%)	n (%)	n (%)
Penicillin V (%)	0	0	0	0	0	0	0	0	0	0	0
Amoxicillin (%)	1	0	0	0	0	0	1 (100)	0	0	0	0
Amoxicillin–clavulanic acid (%)	28	0	0	0	1 (3.6)	0	26 (92.8)	0	0	1 (3.6)	0
Macrolide or clindamycin (%)	1	0	0	0	0	0	1 (100)	0	0	0	0
Cephalosporin (%)	59	1 (1.7)	0	0	8 (13.6)	1 (1.7)	41 (69.5)	0	0	8 (13.6)	0
Fosfomycin (%)	169	14 (8.3)	80 (47.3)	2 (1.2)	4 (2.4)	0	60 (35.5)	2 (1.2)	1 (0.6)	3 (1.8)	1 (0.6)
Nitrofurantoin (%)	29	0	1 (3.4)	0	5 (17.2)	0	16 (55.2)	1 (3.4)	0	5 (17.2)	1 (3.4)
TMP-SMX (%)	13	0	0	0	0	1 (7.7)	8 (61.5)	0	1 (7.7)	2 (15.4)	1 (7.7)
Quinolone (%)	39	1 (2.6)	0	0	2 (5.1)	1 (2.6)	28 (71.8)	0	0	6 (15.4)	1 (2.6)
Other antibiotic (%)	14	0	0	0	0	0	10 (71.4)	0	0	3 (21.4)	0
TOTAL	353	16	81	2	20	3	192	3	2	28	4

n, number; %, percentage; TMP-SMX, trimethoprim + sulfamethoxazole.

## Data Availability

The data presented in this study are available on request from the corresponding author.

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
