# Peer review of "Antimicrobial Agent Use for Urinary Tract Infection in Long-Term Care Facilities in Spain: Results from a Retrospective Analytical Cohort Analysis"

_antibiotics, 2024, doi:10.3390/antibiotics13020152_

Round 1
Reviewer 1 Report
Comments and Suggestions for Authors
Dear authors, I am pleased to have read your manuscript. I think this should be corrected in depth. Some of my suggestions are as follows.
Please check the title after the antimicrobial should write agents or similar words.
In all the graphs, are the values presented as a range or as an exact number? For instance, in Graph 1, do the numbers 200, 150, 100, and 50 represent the number of patients?
In addition, statistical analyses are indispensable for comparisons between groups in all graphs.
Paragraphs 111-115, A total of 46 residents who exhibited neither general nor specific symptoms indicative of UTI were reported. Were these patients positive for ITU using diagnostic methods? How were these patients possible to classify with a diagnosis of UTI, if they had no symptoms or all diagnostic tests?
For Tables 2 and 3, please erase N, number; %, and percentage of footnotes to the table. The tables are clear without this information.
Table 2 adds the p-values to a new column and the statistical analysis information in the footnote.
Please perform a statistical analysis of the data in Table 3.
A total of 118 patients underwent urinary culture. What are the microorganisms isolated? What is the resistance profile of the microorganisms? These data could improve the manuscript, in addition to the discussion regarding the choice of therapy and days of administration of antimicrobials.
The therapy were in all patients monotherapy?
In graph 4 it is necessary to change the style. the new style should allow to see the differences between the groups. Please add asterisks between the antibiotics that differed between the groups. In addition, we have added statistical analysis information in the footnote to the graph.
In the materials and methods section, the authors refer to the study as a cross-sectional report and as part of a before/after intervention study, which does not correspond to the manuscript, please eliminate what refers to an intervention study.
In 4.3. The statistical Analysis was incorrect. The authors also referred to a comparison of prescriptions before and after the audit for each group of professionals. Please check and correct the manuscript accordingly.
Author Response
|
Yes |
Can be improved |
Must be improved |
Not applicable |
|
|
Does the introduction provide sufficient background and include all relevant references? |
(x) |
( ) |
( ) |
( ) |
|
Are all the cited references relevant to the research? |
(x) |
( ) |
( ) |
( ) |
|
Is the research design appropriate? |
( ) |
( ) |
(x) |
( ) |
|
Are the methods adequately described? |
( ) |
( ) |
(x) |
( ) |
|
Are the results clearly presented? |
( ) |
( ) |
(x) |
( ) |
|
Are the conclusions supported by the results? |
( ) |
(x) |
( ) |
( ) |
Reviewer 1
(x) I would not like to sign my review report
( ) I would like to sign my review report Quality of English Language
(x) I am not qualified to assess the quality of English in this paper
( ) English very difficult to understand/incomprehensible
( ) Extensive editing of English language required
( ) Moderate editing of English language required
( ) Minor editing of English language required
( ) English language fine. No issues detected
Comments and Suggestions for Authors
Dear authors, I am pleased to have read your manuscript. I think this should be corrected in depth. Some of my suggestions are as follows.
1.Please check the title after the antimicrobial should write agents or similar words. ✅
- Antimicrobial agents use for urinary tract infection in long-term care facilities in Spain: Baseline results from a registration audit.
2.In all the graphs, are the values presented as a range or as an exact number? For instance, in Graph 1, do the numbers 200, 150, 100, and 50 represent the number of patients? ✅
- As shown in Graph 1, when cross-referencing variables between general symptoms and those specific to UTIs, the most prominent circles converged at the intersection of confusion with foul-smelling and cloudy urine, indicating the highest frequency of recorded symptoms. These numbers represent exact numbers of patients.
3.In addition, statistical analyses are indispensable for comparisons between groups in all graphs ✅
- Graph 1 lacks the basis for applying statistical significance as it does not involve a comparison.
- In Graph 2, no statistically significant differences were observed among the three groups, likely due to the limited number of pyelonephritis cases (n=14). However, when comparing cystitis with the absence of both (cystitis and pyelonephritis), a correlation is evident (R2=0.94), indicating that the symptom percentages for these two diagnoses are nearly identical.
- Graph 3 show no significant differences, aligning with the previously mentioned constraint of a small sample size for pyelonephritis cases (only 14 instances).
- Paragraphs 111-115, A total of 46 residents who exhibited neither general nor specific symptoms indicative of UTI were reported. Were these patients positive for ITU using diagnostic methods? How were these patients possible to classify with a diagnosis of UTI, if they had no symptoms or all diagnostic tests? ✅
- A total of 46 residents who exhibited neither general nor specific symptoms indicative of a UTI were reported, as they were prescribed antibiotic for this infection. Within this cohort, 43 persons (93.5%) received antibiotic treatment, predominantly with fosfomycin for 1 or 2 days. Out of the 43 treated cases, urine dipstick diagnostic tests were conducted in 65% of instances, while urine cultures were carried out in 35%.
- For Tables 2, please erase N, number; %, and percentage of footnotes to the table. The tables are clear without this information. ✅
- Table 2 and 3 adds the p-values to a new column and the statistical analysis information in the footnote. ✅
- Please perform a statistical analysis of the data in Table 3. ✅
No statistically significant differences were observed: in patients without catheterization, the p value was 0.1, and with urinary catheterization, the p value was 0.3. (Table 2).
Table 2. Distribution of different urinary tract infections according to sex and whether they have a urinary catheter.
|
Diagnosis |
Without urinary catheter (n=677) |
|
With urinary catheter (n=42) |
|
|
||
|
Men |
Women |
|
Men |
Women |
|
Total |
|
|
(n=157) |
(n=520) |
p |
(n=26) |
(n=16) |
p |
(n=719) |
|
|
n (%) |
n (%) |
|
n (%) |
n (%) |
|
n (%) |
|
|
Cystitis |
101 (64.3) |
374 (71.9) |
0.1 |
13 (50.0) |
10 (62.5) |
0.3 |
498 (69.3) |
|
Pyelonephritis |
3 (1.9) |
10 (1.9) |
0.1 |
0 (0.0) |
1 (6.25) |
0.3 |
14 (2) |
|
None of the above |
35 (22.3) |
100 (19.2) |
0.1 |
12 (46.2) |
4 (25.0) |
0.3 |
151 (21.0) |
|
Missing |
18 (11.5) |
36 (6.9) |
0.1 |
1 (3.8) |
1 (6.25) |
0.3 |
56 (7.7) |
No statistically significant differences were observed.
Table 3. Diagnosis and Treatment according by sex
|
|
Total |
Women |
Men |
|
|
|
|
|
n (%) |
n (%) |
n (%) |
|
|
|||
|
|
|
719 (100) |
536 (74.5) |
183 (25.5) |
p |
|
|
|
Diagnosis |
Cystitis (%) |
498 (69.3) |
384 (71.6) |
114 (62.2) |
0.1 |
|
|
|
Pyelonephritis (%) |
14 (1.9) |
11 (2.0) |
3 (1.6) |
0.7 |
|
||
|
None of the above (%) |
151 (21.0) |
104 (19.4) |
47 (25.6) |
0.7 |
|
||
|
Antibiotic dispensed |
Penicillin V (%) |
0 (0.0) |
0 (0.0) |
0 (0.0) |
0.7 |
|
|
|
Amoxicillin (%) |
3 (0.4) |
2 (0.37) |
1 (0.5) |
0.7 |
|
||
|
Amoxicillin/clavulanic acid (%) |
65 (9.0) |
45 (8.3) |
20 (10.9) |
0.3 |
|
||
|
Macrolide or clindamycin (%) |
1 (0.1) |
1 (0.2) |
0 (0.0) |
0.5 |
|
||
|
Cephalosporin (%) |
129 (17.9) |
92 (17.1) |
37 (20.2) |
0.3 |
|
||
|
Fosfomycin (%) |
288 (40.1) |
228 (42.5) |
60 (32.7) |
0.2 |
|
||
|
Nitrofurantoin (%) |
54 (7.5) |
42 (7.8) |
12 (6.5) |
0.5 |
|
||
|
Trimethoprim/sulfamethoxazole (%) |
34 (4.7) |
19 (3.5) |
15 (8.2) |
0.1 |
|
||
|
Quinolone (%) |
86 (12.0) |
59 (11.0) |
27 (14.7) |
0.1 |
|
||
|
Other antibiotic (%) |
26 (3.6) |
19 (3.5) |
7 (3.8) |
0.8 |
|
||
|
No antibiotic (%) |
40 (5.6) |
32 (5.9) |
8 (4.3) |
0.4 |
|
||
No statistically significant differences were observed.
- A total of 118 patients underwent urinary culture. What are the microorganisms isolated? What is the resistance profile of the microorganisms? These data could improve the manuscript, in addition to the discussion regarding the choice of therapy and days of administration of antimicrobials. ✅
- Lines 153-157: Concerning the diagnosis, cystitis was the most frequently recorded, accounting for almost 70% of the patients; however, only 15% of the patients undertook a urine culture. Unfortunately, results on the isolated microorganisms and their resistance profile were not available.
- The therapy were in all patients monotherapy? ✅
- All included cases received antibiotic monotherapy.
- In graph 4 it is necessary to change the style. the new style should allow to see the differences between the groups. Please add asterisks between the antibiotics that differed between the groups. In addition, we have added statistical analysis information in the footnote to the graph. ✅
- We describe the information in graph 4 in a paragraph: When studying antibiotic prescriptions, differences were identified in the prescribing patterns for pyelonephritis in comparison to acute cystitis. Specifically, there was a higher prevalence of quinolones and cephalosporins in pyelonephritis cases, while fosfomycin was less frequently prescribed compared to cases of acute cystitis. No statistically significant differences were observed among the three groups.
- In the materials and methods section, the authors refer to the study as a cross-sectional report and as part of a before/after intervention study, which does not correspond to the manuscript, please eliminate what refers to an intervention study. ✅
- This is a cross-sectional study.
- In 4.3. The statistical Analysis was incorrect. The authors also referred to a comparison of prescriptions before and after the audit for each group of professionals. Please check and correct the manuscript accordingly. ✅
- We used the Chi-square test to the frequencies of prescriptions after and before the audit for each group of professionals and setting/country, to test the null hypothesis of no effect of the audit. Antibiotic prescription was considered as the dependent variable (yes/no). Those models control for symptoms and duration, as well as for the demand for antibiotics by the resident. The goodness of fit was assessed using the Wald test of the model, with the deviance test to compare alternative models. Statistical significance was considered with P<0.05. The data were analyzed with the Stata v17 statistical program.

Reviewer 2 Report
Comments and Suggestions for Authors
Thanks for giving me an opportunity to review.
-Paper is very well detailed and comprehensive.
- I like the idea of interaction charts.
Graph 2: Did pyelonephritis present asymptomatically in 40 individuals. How common is this?
Table 2/ Table 3: Add p-value as a separate column.
How do the authors plan to balance the need of targeted antibiotic use only possible with a urine culture with the need to quickly control UTIs in a vulnerable geriatric population? Can the authors suggest some guidelines.
Line 354: Chisq could be used if the cell counts are >5. It is more robust than Fisher's.
Author Response
Reviewer 2
( ) I would not like to sign my review report
(x) I would like to sign my review report
Quality of English Language
( ) I am not qualified to assess the quality of English in this paper
( ) English very difficult to understand/incomprehensible
( ) Extensive editing of English language required
( ) Moderate editing of English language required
( ) Minor editing of English language required
(x) English language fine. No issues detected
|
Yes |
Can be improved |
Must be improved |
Not applicable |
|
|
Does the introduction provide sufficient background and include all relevant references? |
(x) |
( ) |
( ) |
( ) |
|
Are all the cited references relevant to the research? |
(x) |
( ) |
( ) |
( ) |
|
Is the research design appropriate? |
(x) |
( ) |
( ) |
( ) |
|
Are the methods adequately described? |
(x) |
( ) |
( ) |
( ) |
|
Are the results clearly presented? |
(x) |
( ) |
( ) |
( ) |
|
Are the conclusions supported by the results? |
(x) |
( ) |
( ) |
( ) |
Comments and Suggestions for Authors
Thanks for giving me an opportunity to review.
-Paper is very well detailed and comprehensive.
- I like the idea of interaction charts.
- Graph 2: Did pyelonephritis present asymptomatically in 40 individuals. How common is this? ✅
- Included section 3.2. Lines 233-237.
- As for the 46 patients that were asymptomatic and were treated with antibiotics, there is limited direct evidence concerning cases where antibiotics are administered specifically to asymptomatic patients within this population group. It is crucial for future research to concentrate on comprehending the factors contributing to antibiotic administration despite the absence of symptoms in patients.
- Table 2/ Table 3: Add p-value as a separate column. ✅
Table 2. Distribution of different urinary tract infections according to sex and whether they have a urinary catheter.
|
Diagnosis |
Without urinary catheter (n=677) |
|
With urinary catheter (n=42) |
|
|
||
|
Men |
Women |
|
Men |
Women |
|
Total |
|
|
(n=157) |
(n=520) |
p |
(n=26) |
(n=16) |
p |
(n=719) |
|
|
n (%) |
n (%) |
|
n (%) |
n (%) |
|
n (%) |
|
|
Cystitis |
101 (64.3) |
374 (71.9) |
0.1 |
13 (50.0) |
10 (62.5) |
0.3 |
498 (69.3) |
|
Pyelonephritis |
3 (1.9) |
10 (1.9) |
0.1 |
0 (0.0) |
1 (6.25) |
0.3 |
14 (2) |
|
None of the above |
35 (22.3) |
100 (19.2) |
0.1 |
12 (46.2) |
4 (25.0) |
0.3 |
151 (21.0) |
|
Missing |
18 (11.5) |
36 (6.9) |
0.1 |
1 (3.8) |
1 (6.25) |
0.3 |
56 (7.7) |
No statistically significant differences were observed.
Table 3. Diagnosis and Treatment according by sex
|
|
Total |
Women |
Men |
|
|
|
|
|
n (%) |
n (%) |
n (%) |
|
|
|||
|
|
|
719 (100) |
536 (74.5) |
183 (25.5) |
p |
|
|
|
Diagnosis |
Cystitis (%) |
498 (69.3) |
384 (71.6) |
114 (62.2) |
0.1 |
|
|
|
Pyelonephritis (%) |
14 (1.9) |
11 (2.0) |
3 (1.6) |
0.7 |
|
||
|
None of the above (%) |
151 (21.0) |
104 (19.4) |
47 (25.6) |
0.7 |
|
||
|
Antibiotic dispensed |
Penicillin V (%) |
0 (0.0) |
0 (0.0) |
0 (0.0) |
0.7 |
|
|
|
Amoxicillin (%) |
3 (0.4) |
2 (0.37) |
1 (0.5) |
0.7 |
|
||
|
Amoxicillin/clavulanic acid (%) |
65 (9.0) |
45 (8.3) |
20 (10.9) |
0.3 |
|
||
|
Macrolide or clindamycin (%) |
1 (0.1) |
1 (0.2) |
0 (0.0) |
0.5 |
|
||
|
Cephalosporin (%) |
129 (17.9) |
92 (17.1) |
37 (20.2) |
0.3 |
|
||
|
Fosfomycin (%) |
288 (40.1) |
228 (42.5) |
60 (32.7) |
0.2 |
|
||
|
Nitrofurantoin (%) |
54 (7.5) |
42 (7.8) |
12 (6.5) |
0.5 |
|
||
|
Trimethoprim/sulfamethoxazole (%) |
34 (4.7) |
19 (3.5) |
15 (8.2) |
0.1 |
|
||
|
Quinolone (%) |
86 (12.0) |
59 (11.0) |
27 (14.7) |
0.1 |
|
||
|
Other antibiotic (%) |
26 (3.6) |
19 (3.5) |
7 (3.8) |
0.8 |
|
||
|
No antibiotic (%) |
40 (5.6) |
32 (5.9) |
8 (4.3) |
0.4 |
|
||
No statistically significant differences were observed.
- 3. How do the authors plan to balance the need of targeted antibiotic use only possible with a urine culture with the need to quickly control UTIs in a vulnerable geriatric population? Can the authors suggest some guidelines.
- Given the high incidence of antibiotic prescription in this population, which can be both ineffective and potentially harmful, the research group recommended a treatment protocol aligned with the guidelines outlined by Gupta et al and Bonkat et al. [46,47]
- Line 354: Chiq could be used if the cell counts are >5. It is more robust than Fisher's.
- We used the Chi-square test to the frequencies of prescriptions after and before the audit for each group of professionals and setting/country, to test the null hypothesis of no effect of the audit. Antibiotic prescription was considered as the dependent variable (yes/no). Those models control for symptoms and duration, as well as for the demand for antibiotics by the resident. The goodness of fit was assessed using the Wald test of the model, with the deviance test to compare alternative models. Statistical significance was considered with P<0.05. The data were analyzed with the Stata v17 statistical program.

Reviewer 3 Report
Comments and Suggestions for Authors
In this manuscript, the authors analyzed the use of various classes of antibiotics for UTI's in NH facilities in Spain. They looked at variants like gender, symptoms, method of detection, class of drugs, and duration of treatment. Overall, this study highlights the need of proper diagnostics followed by better tailored treatment in long-term care facilities.
The strength of this paper is the introduction and discussion. Both the sections are very well written. The graphs are well illustrated and self explanatory. Overall representation of the tables can be improved.
Lines 144-148: Is there any explanation to these findings? Has this pattern been observed in studies elsewhere?
Line 158: Table 3: It is unclear whether the data for antibiotic dispensed is for overall diagnosis or specific to cystitis/pyelonephritis etc. The percentages seem incorrect ...for example, 2 (66,6) in women for Amoxycillin. Please elaborate against what the % is being calculated.
Line 161-163: please rephrase. Unclear what the authors want to convey.
Overall, based on the the results discussed in Lines 161-168, what were the impacts/outcomes for this pattern of prescription? Was that analyzed under the scope of this paper?
Lastly, are there any studies that stress on the observation of symptoms of change in mental status/ confusion for younger patients? Is there any co-relation of age and symptoms observed for UTI. Please include in discussion, if any.
Author Response
|
Yes |
Can be improved |
Must be improved |
Not applicable |
|
|
Does the introduction provide sufficient background and include all relevant references? |
(x) |
( ) |
( ) |
( ) |
|
Are all the cited references relevant to the research? |
(x) |
( ) |
( ) |
( ) |
|
Is the research design appropriate? |
( ) |
( ) |
(x) |
( ) |
|
Are the methods adequately described? |
( ) |
( ) |
(x) |
( ) |
|
Are the results clearly presented? |
( ) |
( ) |
(x) |
( ) |
|
Are the conclusions supported by the results? |
( ) |
(x) |
( ) |
( ) |
Reviewer 3
( ) I would not like to sign my review report
(x) I would like to sign my review report Quality of English Language
( ) I am not qualified to assess the quality of English in this paper
( ) English very difficult to understand/incomprehensible
( ) Extensive editing of English language required
( ) Moderate editing of English language required
( ) Minor editing of English language required
(x) English language fine. No issues detected
|
Yes |
Can be improved |
Must be improved |
Not applicable |
|
|
Does the introduction provide sufficient background and include all relevant references? |
(x) |
( ) |
( ) |
( ) |
|
Are all the cited references relevant to the research? |
(x) |
( ) |
( ) |
( ) |
|
Is the research design appropriate? |
( ) |
(x) |
( ) |
( ) |
|
Are the methods adequately described? |
(x) |
( ) |
( ) |
( ) |
|
Are the results clearly presented? |
(x) |
( ) |
( ) |
( ) |
|
Are the conclusions supported by the results? |
( ) |
(x) |
( ) |
( ) |
Comments and Suggestions for Authors
In this manuscript, the authors analyzed the use of various classes of antibiotics for UTI's in NH facilities in Spain. They looked at variants like gender, symptoms, method of detection, class of drugs, and duration of treatment. Overall, this study highlights the need of proper diagnostics followed by better tailored treatment in long-term care facilities.
The strength of this paper is the introduction and discussion. Both the sections are very well written. The graphs are well illustrated and self explanatory. Overall representation of the tables can be improved.
1.Lines 144-148: Is there any explanation to these findings? Has this pattern been observed in studies elsewhere? ✅
- There is limited direct evidence addressing cases where antibiotics are not administered specifically within this population group. It is important that future research focuses on understanding factors contributing to non-treatment with antibiotics despite confirmed UTI diagnoses among elderly residents living in nursing.
- Line 158: Table 3: It is unclear whether the data for antibiotic dispensed is for overall diagnosis or specific to cystitis/pyelonephritis etc. The percentages seem incorrect ...for example, 2 (66,6) in women for Amoxycillin. Please elaborate against what the % is being calculated.
Table 3. Diagnosis and Treatment according by sex
|
|
Total |
Women |
Men |
|
|
|
|
|
n (%) |
n (%) |
n (%) |
|
|
|||
|
|
|
719 (100) |
536 (74.5) |
183 (25.5) |
p |
|
|
|
Diagnosis |
Cystitis (%) |
498 (69.3) |
384 (71.6) |
114 (62.2) |
0.1 |
|
|
|
Pyelonephritis (%) |
14 (1.9) |
11 (2.0) |
3 (1.6) |
0.7 |
|
||
|
None of the above (%) |
151 (21.0) |
104 (19.4) |
47 (25.6) |
0.7 |
|
||
|
Antibiotic dispensed |
Penicillin V (%) |
0 (0.0) |
0 (0.0) |
0 (0.0) |
0.7 |
|
|
|
Amoxicillin (%) |
3 (0.4) |
2 (0.37) |
1 (0.5) |
0.7 |
|
||
|
Amoxicillin/clavulanic acid (%) |
65 (9.0) |
45 (8.3) |
20 (10.9) |
0.3 |
|
||
|
Macrolide or clindamycin (%) |
1 (0.1) |
1 (0.2) |
0 (0.0) |
0.5 |
|
||
|
Cephalosporin (%) |
129 (17.9) |
92 (17.1) |
37 (20.2) |
0.3 |
|
||
|
Fosfomycin (%) |
288 (40.1) |
228 (42.5) |
60 (32.7) |
0.2 |
|
||
|
Nitrofurantoin (%) |
54 (7.5) |
42 (7.8) |
12 (6.5) |
0.5 |
|
||
|
Trimethoprim/sulfamethoxazole (%) |
34 (4.7) |
19 (3.5) |
15 (8.2) |
0.1 |
|
||
|
Quinolone (%) |
86 (12.0) |
59 (11.0) |
27 (14.7) |
0.1 |
|
||
|
Other antibiotic (%) |
26 (3.6) |
19 (3.5) |
7 (3.8) |
0.8 |
|
||
|
No antibiotic (%) |
40 (5.6) |
32 (5.9) |
8 (4.3) |
0.4 |
|
||
3.Line 161-163: please rephrase. Unclear what the authors want to convey. ✅
- When studying antibiotic prescriptions, differences were identified in the prescribing patterns for pyelonephritis in comparison to acute cystitis. Specifically, there was a higher prevalence of quinolones and cephalosporins in pyelonephritis cases, while fosfomycin was less frequently prescribed compared to cases of acute cystitis. No statistically significant differences were observed among the three groups.
4.Overall, based on the results discussed in Lines 161-168, what were the impacts/outcomes for this pattern of prescription? Was that analyzed under the scope of this paper? ✅
- We have conducted a new search, but the studies in the literature do not specify whether they address cystitis or pyelonephritis; rather, they do not provide information on urinary tract infections in general.
- Lastly, are there any studies that stress on the observation of symptoms of change in mental status/ confusion for younger patients? Is there any co-relation of age and symptoms observed for UTI. Please include in discussion, if any. ✅
- In the literature, I have identified a correlation between UTIs and symptoms of altered mental status/confusion in both pediatric and geriatric populations. While clinical practice recognizes that patients with neurological issues, such as stroke or intracranial hemorrhage, may exhibit similar symptoms, this association is not extensively documented in the literature.
- Lines 215-222: Caterino et al. established a correlation between age and UTI symptoms, revealing that a significant proportion of adults aged 65 or older diagnosed with UTIs in the emergency department lacked fever and typical urinary tract symptoms. In this age group, individuals are less prone to exhibit urinary tract symptoms compared to their younger counterparts. Notably, those aged 85 and older are more likely to manifest altered mental status. Additionally, nursing home residents are characterized by a reduced likelihood of urinary tract symptoms but an increased likelihood of fever and altered mental status.

Round 2
Reviewer 1 Report
Comments and Suggestions for Authors
The manuscript has been improved accordingly. Please check point 2 of my review above.
2. In all the graphs, are the values presented as a range or as an exact number? For instance, in Graph 1, do the numbers 200, 150, 100, and 50 represent the number of patients?
You mentioned that it represents the exact number of patients.
In line 115, 46 residents exhibited neither general nor specific symptoms. In Graph 1, the circle indicates 50 patients.
Author Response
Open Review
Comments and Suggestions for Authors
The manuscript has been improved accordingly. Please check point 2 of my review above.
- In all the graphs, are the values presented as a range or as an exact number? For instance, in Graph 1, do the numbers 200, 150, 100, and 50 represent the number of patients?
You mentioned that it represents the exact number of patients.
In line 115, 46 residents exhibited neither general nor specific symptoms. In Graph 1, the circle indicates 50 patients.
Answer from authors: Thank you very much for your comments that help us improve the manuscript.
The ball chart is according to "n observations". It is normal that it is difficult to appreciate a difference between 46-50, since they are very close values, but the ball of "None of the specifics - No general symptoms" is smaller than the one in the Legend that indicates 50. If you take a ruler, the ball of 50 has a diameter of 5 millimeters, while the ball of 46 does not reach 5 millimeters. It is very similar as usual and hard to appreciate, but in a range over 200 the difference between 46-50 is less than 2%.
I add a word so the reviewer can see the graph
